# The Lockean Proviso and Orbital Sustainability— An Anthropological View

**Lucian Mocrei-Rebrean**

Department of Humanities and Social-Political Sciences, Stefan cel Mare University, 720229 Suceava, Romania; mocreilucian@gmail.com

**Abstract:** Over the last decades, we have witnessed the gradual commercialization of the Earth orbit. The exponential development of private space activities makes this distant natural field, with the overcoming of technological difficulties, more and more hospitable to free initiative and entrepreneurship. However, the orbital space is considered global commons. Through the imaginary case method, we intend to ponder on possible ways to legally regulate the exploitation of the orbital space, namely the application of Pigouvian taxes, on the sustainability of the orbital environment, through ethical considerations originating from the application of the Lockean proviso. Although they are designed to cover the damage caused by that particular polluting activity, which is difficult to estimate and, in our case, almost impossible to quantify in the long run, the Pigouvian taxes are the result of a proactive logic. The tension between civilization and nature turns the world outside the Earth into a wilderness destined for humanization, another area of exercise of the liberal self. Non-legal reasons for the sustainability of the orbital environment may arise from observing the Lockean principle of fair ownership. Between the prohibition of an unreasonable destruction of nature's goods and the equitable access to extra-terrestrial resources, the human desire for appropriation updates the proviso destined for the colonization of America in the twenty-first century. Given that there are currently no plans to clean the technological waste in orbit, adopting the conservation of the orbital environment as an ethical principle could help to formulate a more environmentally responsible liberalism, as part of a long-term agenda of exploitation in the vicinity of our planet.

**Keywords:** orbital environment; pollution; sustainability; Pigouvian taxes; conservation of resources; Lockean proviso

## 1. Introduction

Moral positioning regarding environmental issues is becoming increasingly relevant in public discourse. As an argument, moral reasoning has always had an impact on political decisions, helping to legitimize medium and long-term strategies. The awareness of a shared vulnerability faced with global environmental hazards can become a strong motivating factor in promoting the pro-environmental behavior, as well as in favoring and implementing environmental policies [1].

Given the actual magnitude of the pollution generated by technological activities in natural environments, not taking into account the ethical aspects in the political decision-making process can jeopardize the legitimate interests of future generations on a global scale. Therefore, focusing on the impact of moral sensitivity on environmental policies is a topic of current interest. In this respect, the adoption of international laws needs a common moral ground, in order to enforce rules and regulations for the long-term protection of the global environment. However, an eventual synchronization of national policies seems to remain a long-term intergenerational goal [2].

Human exploration, as well as commercial and military activity outside the Earth's atmosphere are growing in scale and intensity with each passing decade. Drawn by convention and constantly overtaken, the famous Kármán line appears less and less to

us as an immutable frontier. That hostile and anonymous natural wilderness beyond it receives identity and value as it is humanized through knowledge and action. Kanon argues that exploring and exploiting its resources may become necessary in order to ensure the long-term sustainability of humanity's vital resources [3]. Given that the sustainability of the terrestrial environment is now threatened by global pollution, it is possible that the survival of humanity will depend on the exploitation of extra-terrestrial resources.

Rosenberg points out that we have been witnessing, for several decades now what might be called a de facto colonization of Earth's orbit. Any colonization has long-term effects on the environment, as has happened whenever mankind has spread to new continents [4]. Over the last decades, we have witnessed the gradual commercialization of Earth's orbit. The exponential development of private space activities makes this distant natural field, with the overcoming of technological difficulties, more and more hospitable to free initiative and entrepreneurship [5]. In the context of the current trend of re-launching the space industry, in addition to civilian space systems and military equipment already present in the Earth orbit for many decades, we are witnessing an unprecedented intensification of commercial activity, especially in Low-Earth Orbit. New business opportunities are emerging in the field of global internet services. Further developments such as the future creation of in-situ orbital conveniences would be extremely profitable for low-latency data providers [6].

Ongoing projects such as telecommunications satellite networks, which aim to provide global connectivity services and for which ongoing in-orbit maintenance becomes a condition of feasibility, pose increasingly serious problems related to the long-term sustainability of the orbital environment. Access to outer space can bring huge economic benefits. From this point of view, the immediately relevant aspect of human activity in space is the commercial exploitation of Earth's orbit. As private initiative in Earth's orbit cannot currently be legally controlled strictly enough, the orbital environment can become a completely unrestricted area of freedom of action.

## 2. Literature Review

Although still in an early phase, the concern about the exploitation of extraterrestrial natural resources is increasingly emphasized upon when it comes to environmental sustainability. There are contributions from proactive, anticipatory authors coming from the applied ethics area or from case law, even if, given the relative novelty of the debate, they are not very thorough. These contributions remain valuable because they are critically positioned in a constructive way in relation to current and future environmental policies. We will present briefly some bibliographical landmarks, in order to familiarize the non-specialized reader with the topic of our debate.

Given that global sustainability is increasingly threatened by pollution and that the coordination of national policies in a timely manner remains difficult, Kanon's position is of particular relevance. It claims that, due to technological advances, the prospecting and exploitation of resources outside the Earth should be taken into account in order to safeguard the common interest: the sustainability of the vital resources of a humanity that consumes more and more of the surrounding terrestrial nature. It claims that our survival is likely to depend, in the not too distant future, on the intensive exploitation of alien resources, as a solution to protect the terrestrial environment [3].

Regarding the issue of ensuring the sustainability of the, increasingly exploited, orbital environment and starting from the finding that private entities do not invest on their own initiative in environmental risk management, Oz, Bullock and Johanson propose proactive solutions based on legal analogies. Oz states that the legal regime for the exploitation of riparian basins, a common local resource, currently regulated in such a way that it does not jeopardize the interest of future generations, be adapted and applied to the orbital environment—a common global resource [7]. Alternatively, Bullock and Johanson advocate for the management of environmental risks assumed by the trade exploitation of the orbit through the financial stimulation of private operators to invest in sustainability [8].

From the perspective of environmental ethics, the issue of the sustainability of the orbital environment falls under a pro-environmentalist discourse that has constantly pressed environmental policies since the middle of the 20th century. The problem of intergenerational justice remains as current in the new historical conjuncture. Attfield, for example, insists that the conservation of natural resources on a global scale is a moral imperative because current generations must leave behind a world whose capacity of ensuring vital needs is not reduced in any way [9].

Given the constant interest, growing exponentially, of the private entrepreneur for the exploitation of the orbital environment, authors such as Rosenberg and Pilchman anticipate that, given the alert rhythm of the advancement of space technologies, asteroid mining or helium mining from the Moon are quite plausible scenarios to be considered. This draws attention to risks of an ethical nature. Rosenberg states that we are witnessing a de facto colonization of the orbit with long-term consequences on the global environment. He argues that Locke's eighteenth-century reflections are of surprising relevance [4]. Indeed, the Lockean concept of original appropriation allows for consideration of any newly discovered, virgin part of nature through the perspective of a relevant opposition for sustainability issues, that between civilization and wildness. Nature exploitation is not only permitted but even desirable, while economic behavior must always reflect moral sensitivity to the environment. In order to not damage future acts of individual appropriation of a part of extraterrestrial nature, one should observe three ethical requirements synthesized in what is known as the Lockean proviso. Pilchman argues that the logic of the original appropriation allows for the formulation of moral objections against the possibility of a legally unrestricted exploitation of the extraterrestrial environment [10].

## 3. Methodology

We intend to ponder on possible ways to legally regulate commercial operations in orbit, namely the application of Pigouvian taxes on the sustainability of the orbital environment, through ethical considerations that originate from the application of the Lockean proviso.

The need for a more proactive way of thinking in tackling global environmental issues makes moral intuitions become more relevant [1]. It is a matter of responsible anticipation, hence the relevance of the imaginary case method. Heuristic arguments can be useful conceptual tools in formulating hypotheses or establishing conjectures in an investigation. Imaginary cases especially have a distinct role in applied ethics, because they can lead to the discovery of important aspects from which we can infer what may be important in other similar situations. They contribute to the process of moral knowledge by allowing us to consider not only a list of things that might matter, but also how they might matter [11]. Adding imaginary cases to a collection of existing cases can highlight patterns or structures in our moral judgments. Thus, although the judgment required is most often intuitive, imaginary experiments have not only a heuristic utility, but they also prepare us effectively for the choice of a particular course of action in real situations. Because the lines of reasoning always extend beyond the cases that are actually taken into debate, imaginary experiments can help us discover moral judgments relevant to what we might do in the future [12].

What enables us to learn from a succession of what we can consider morally significant cases is reasoning by analogy [12]. Its premise is the finding that the experience, always updated, of new and newly lived cases develops our moral sensitivity, serving as a guide for action for the future [13]. In these circumstances, we must also consider the limits of reasoning by analogy. We build imaginary cases to increase our ability to deliberate in real cases. Dancy notices that the way we construct these always depends directly on that already existing knowledge, knowledge acquired from the experience of previous, real, already consumed cases [12]. Therefore, it remains difficult to conclusively assess situations that are not yet real and even imaginary variants of real situations.

In light of the above considerations, how relevant an analogy is will depend on the extent to which real and imagined situations differ. The more elements the imagined case

has in common with the real ones, the stronger our analogy will be. Analogical reasoning can play an instrumental role in environmental law because it requires the application of rules in new situations. In our case, we could approach the exploitation of Earth's orbit by analogy with the exploitation of international waters. In fact, many articles from the Space Treaty are inspired from The Law of the Sea. We think the analogy is strong enough in terms of the problems posed by environmental sustainability because, like the orbital space, everything outside the territorial waters and areas of economic exclusivity constitutes global commons. Exploiting an orbital altitude as part of a business enterprise is in many ways similar to the transit of a commercial ship through international waters.

There is a second relevant analogy recently proposed by Oz. There should be no legal difference between the river basins operation regime, which is regulated in such a way as not to jeopardize the access of future generations to a common regional resource and the Earth's orbit as a common resource for all mankind [7]. Someone can easily argue that securing the sustainability of orbital space is not as vital as water security, but such an analogy can help us move beyond our current experience and anticipate new situations in light of our current commitments to the terrestrial environment. Any possible knowledge about new cases can only be based on our previous experience. In a debate on ensuring the sustainability of the orbital environment, we can successfully isolate the factors that are morally relevant to our possible engagement, starting from the current debates on environmental ethics in resource exploitation.

### 4. Results and Discussion

*4.1. The Earth's Orbit—Global Commons*

An imaginary case must always be close to a real situation. In our case, an ethical assessment of the commercial activity in Earth's orbit is close enough to any current debate on global ecology and can be taken into account.

Given the current rate of proliferation of orbital debris, future generations' access to the orbit's benefits could suffer. Although the development of an operational strategy for orbital waste management faces serious international economic and political dissensions, amid a resurgence of the arms race, innovative, creative and anticipatory policies are absolutely necessary to encourage the sustainable use of orbital space [8].

There is an international treaty currently in force, which aims to regulate the exploration and exploitation of extra-terrestrial space, including Earth's orbit. It was signed in 1967 and is entitled "Treaty on Principles Governing the Activities of States in the Exploration and Use of Outer Space, including the Moon and Other Celestial Bodies" [14]. A brief analysis of the most relevant articles highlights, from the start, an unresolved contradiction: that between the political ideology of res communes and free private initiative.

Article I declares outer space the "province of all mankind" [14]. Hence, any economic exploration and exploitation beyond the Karman Line should be to the benefit of all nations, whether or not they have the opportunity to participate directly in it. At the same time, however, the extra-terrestrial world can be explored and exploited by any entity, public or private, without restrictions and without discrimination under the conditions set out in Article II: no nation can claim sovereignty over any part of the extra-terrestrial realm, by virtue of the right of first come.

Articles VI, VII and VIII are important because they have normative value. The first generally assigns full legal responsibility to states for the consequences of any action in space, whether the author is a government agency or a private entity registered in their territory. The second and third articles state that jurisdiction over objects launched into space belong exclusively to that state for the entire period of their life. Furthermore, both the state of ownership and the launching state shall be jointly liable to the international community for any damage caused to third parties [14].

In conclusion, although the world outside the Earth is seen as a resource accessible to all, that is it remains, from a legal point of view, under the res communes regime, the normative value of the Treaty is questionable. In practice, the two principles on which it is

based: that of equity in access to natural resources and that of freedom in exploration and exploitation are difficult to correlate [15]. In addition, the Treaty dates back to 1967. Since then, the scale of orbit operations has increased so much that regulations do not effectively cover current needs, nor can they anticipate future ones. The sustainability of the Earth's environment and the security of human action in orbit are already under threat [16]. With the agglomeration of the orbit with space debris, the risks associated with future human exploration missions increase gradually but steadily.

### 4.2. The Pigouvian Taxes and the Orbit as a Natural Environment

Recognizing that responsible environmental action always involves additional costs, Oz notes that private entities are rarely motivated to invest enough on their own initiative to properly manage environmental risks. How, then, can private entities that carry out commercial activities in outer space be held accountable? He states that there should be no legal difference between the river basins operation regime, which is regulated in such a way as not to jeopardize the access of future generations to a common national resource, and the Earth's orbit as a common resource for all mankind [7]. In the same vein, Bullock and Johanson argue that providing financial incentives for investment in sustainability as well as taxation associated with environmental risks can improve the security of the terrestrial environment and increase the security of space launches [8].

Salter examines the appropriateness of these taxes in the exploitation of the orbital environment based on the idea that space activities should fall under private law since they can create negative externalities [17]. Any form of pollution is an externality insofar as it has negative, proven and relatively quantifiable effects on other persons or communities. Governments impose additional costs on any polluting activity by forcing the polluter to pay the equivalent of the harm they cause to the community. Such taxes, called Pigouvian taxes, are the legal expression of a compensatory logic in the sense that they do not expressly prohibit pollution, they just try to discourage it. Although they are designed to cover the damage caused by the polluting activity, damage that is difficult to estimate and, in our case, almost impossible to quantify in the long run, they still show a proactive, anticipatory logic.

As Banzhaf points out, the legitimacy of those taxes is based on the maintenance of common ownership of natural resources, the exploitation of which is absolutely necessary for the development of certain communities. Because they are public goods, no one can be excluded from the benefits in either the production or the consumption phase [18]. Rivalry in the consumption of common natural resources is significantly hampered and environmental risks in the management of these resources are usually managed in public–private partnerships [19]. In this sense, the Earth's orbital space, now perceived as an extremely hospitable space of free initiative, can gradually turn into a multitude of private, temporarily rented altitudes [20]. Even so, owning one is still beyond the reach of the average person. Beyond the legal responsibility there is, assumed or not at the organizational level, a moral responsibility on the part of the private entities that might be currently operating in the orbit of the Earth and profitably using a resource that is, in principle, available to everyone.

Although the orbit is currently being monitored, more or less efficiently, by various government agencies and the responsibility for any damage caused by orbital debris can be proven, the payment of this damage remains at the discretion of the states involved. It cannot be enforced effectively. How, then, can scientific predictions of the risk posed by space debris to human activity in the orbital environment contribute to the regulation of the situation in the short, medium and long term? This is a question that gives rise to ethical considerations related to those public debates on ecology that exert a constant pressure on terrestrial environmental policies, not without results.

Any part of the natural world initially exists as a landscape, it is unaffected by the consequences, whether intentional or not, of human presence. Once they appear, that landscape gradually becomes an environment. Based on this finding, Berleant distinguishes the three ways in which people experience their encounter of the natural world: the contemplative-distant mode, the active-employed mode and, finally, the participatory

mode, which involves action, a direct, sustained relationship, and an approach that can ultimately transform the environment [21]. Thanks to technology, as a region beyond the lower boundary of the Earth's ionosphere, the orbit is increasingly becoming a topos of human action, a natural environment or, as Gorman describes it, an integral part of an ever-expanding anthropic landscape [22], a landscape that can be imagined from launch facilities, continuing outward with the two space stations, thousands of telecommunications and surveillance satellites, with robotic probes on other planets and so on.

Hereinafter, we will try to show that a debate in terms of environmental ethics about the exploitation of the orbit is current and timely.

First, as an environment that increasingly falls under that utilitarian logic that governs human action on Earth, the orbit turns out to be a natural resource intensely exposed to both private competition and international conflict. The common interest, in the broadest sense that can be attributed to this term, is inevitably affected [23]. The danger of an instrumentalization without any restraint of the surrounding nature has always existed. To deny this only because we are referring to a part of it that is so far from our daily experience is not to consider the perpetual dispute between the private and public interest, a dispute that marks any activity of human exploitation of natural resources [24]. Moreover, it means to overlook an ever-present contradiction: that between the proven power of the spirit of free initiative and the generous, but much less practical, notion of res communes, a notion on which the whole statement is based that the orbit is an integral part of our common heritage.

Secondly, moral considerations are also relevant because the intensive orbital exploitation could, in the long run, become sufficiently harmful for the terrestrial environment, that it would require the adoption of restrictive or even prohibitive measures at international level, despite the ever-increasing commercial benefits that it brings [24]. An immediate concern for the conservation of the orbit as an essential natural resource for the future then becomes more than justified [10]. The political challenge that environmental ethics has assumed since the second half of the twentieth century in the context of large-scale industrialization has been to push for the protection of the terrestrial environment. An extension of this concern to the world beyond the Karman line seems absolutely legitimate as current technological capabilities break the boundary between terrestrial and extra-terrestrial nature.

Thirdly, paying more attention to the ethical aspects of space exploration and exploitation can make a significant contribution to clarifying some legal concepts, not yet sufficiently precise, that are used in formulating regulations for operations in orbit and even on the ethical basis of human activity in other planetary environments that are being explored [25].

In conclusion, given the unpredictability of exploration and exploitation activities in unknown environments, it becomes necessary to reflect proactively on human action in space and from the point of view of environmental ethics, a reflection that can balance the economic or political conflicts involved by the exploitation of resources [26].

*4.3. The Orbital Environment as a Natural Wilderness*

Starting from a well-known distinction, the one proposed by Callicott and Mumford, depending on the importance given to human needs and interests in relation to the integrity and well-being of nature, there are two alternatives in relation to the environment: conservation or preservation [27]. If we recognize the value of nature regardless of the benefits that its exploitation offers then there are reasons to talk about preservation. If we consider that nature is valuable only as a resource, appreciating it only in terms of our needs and priorities, then it must be protected from abuse in the common interest or in the interest of future generations. In this case, we are talking about conservation. This anthropocentric view of the instrumental value of natural items justifies the exploitation of resources in terrestrial environments. As the only valorizing agents, we automatically confer value on nature only by entering into a relationship with it, during our historical

action [28]. Assessed in these terms, the world outside the Earth's atmosphere has an enormous economic value through the invaluable wealth of mineral resources, insofar as technology will allow their exploitation.

In Locke's classic anthropocentric vision, currently practiced by Hardin, among others, nature is, from the beginning, destined for humanization [29]. In this sense, although it remains at a great distance from our daily life, the Earth's orbit can be considered as one of those hostile parts of nature in the process of domestication, which serves the needs of human civilization. The Lockean concept of natural wilderness is, as Mexal points out, linked to the description of the natural world, through the prism of a founding opposition, that between civilization and savagery [15]. The meaning of the concept of savagery is closely linked to Locke's early account of liberal individualism. Imagining an initial state of nature, a primum naturae, Locke locates in it the liberal self, theologically defining the human condition. Pinnacle of God's creation, man is part of nature, but unlike all other creatures, he is superior to all of them. His relationship with nature has to do both with his own individual, practical reason, and divine revelation. Although natural and revealed knowledge are different, they cannot contradict each other: the first confirms the second by emphasizing a fundamental human right, the right to self-preservation. All the products of nature are available to man primarily to ensure his subsistence, but also to improve their living conditions [30].

Justified by legitimate needs, human rights over the natural world are not in perpetuity. The absolute ownership of creation can only belong to the Creator, man has only the freedom to use it, to dispose freely, for the limited duration of his life, of the goods that nature offers him with incessant generosity. The fact that the rights of the natural world are common, granted in advance to all past, present, and future humanity makes them inalienable to all individuals who have existed, exist, or will exist throughout history [30]. The individual effort of self-actualization through work in the midst of the abundance of a nature whose value we discover and amplify through our active presence is defining for the human condition. This virgin nature is not totally deprived of value until the coming of man, but only human work can actually value it. Not the passivity of contemplation but the courageous private initiative, the ability to act creatively and ever-persevering effort can gradually imprint a rational order on the surrounding world which becomes, par excellence, the field of exercise of the liberal self. Throughout this process, that initial value, which the spontaneous abundance of nature already has, is increased. Only the assiduous work of extracting and processing raw materials can bring a most necessary added value [30]. Man's vocation is to become the owner of that part of nature that he can gradually acquire through hard work.

It is obvious that the Lockean notion of natural wilderness has anthropological roots. Any part of the natural world whose abundance lies unused is a wilderness that awaits the transformative touch of man. By effectively serving human needs, nature, whose ultimate reason to be is humanity's well-being, becomes infinitely more precious than if it were ignored, left in its original state. That is why the vocation of the individual is to conquer the common wilderness as it becomes accessible to him [30].

At the beginning of history, the whole world was like America before colonization. By this paradigmatic statement, Locke does not necessarily intend to evoke a concrete situation, but to describe by comparison that initial state of human existence to which his whole argument refers in order to justify, theologically as well, the colonization of the American continent. Although human existence in this original virgin wilderness, in this world of the beginning, presupposed a seemingly unlimited freedom to dispose of one's own person or property, it does not allow self-destruction, nor does it grant man the right to destroy nature [30]. This last statement is of particular importance because the Earth's orbit becomes the vast space for the development of entrepreneurial freedom, with all the environmental risks that it entails, as seen above.

According to Locke, the existence of civilization always implies a nature that remains constantly outside of it. The savagery begins where human order fades long enough for

man to notice its lack. Civilization then means the transformation of any part of newly discovered nature into that world governed by the ethical rules that ownership requires. Assuming the status of "civilized" puts humanity in a presumably constructive tension with any new savagery. It becomes imperative that its natural goods, its spontaneous products, which already belong to all of humanity and do not yet benefit anyone in particular, be exploited for the benefit of all. In the context of this desideratum, an economic expansion, no matter how aggressive, of human civilization beyond the Karman line is absolutely justifiable, regardless of the possible environmental risks that it would entail.

According to Locke's vision, the privatization of the world outside of Earth, starting with the Earth's orbit, appears as a natural right, guaranteed to any socio-economic entity that has the necessary technological means. This kind of liberal mindset may raise legitimate concerns because the exploitation of new natural resources is absolutely justified. The exponentially growing demands of ever-expanding consumer societies [31] are making geological prospecting initiatives beyond the Karman line to be taken more and more seriously by wealthy individuals.

Unlimited natural wealth has been waiting for eons to be appropriated. The world outside the Earth can at any time become the virgin realm of unprecedented entrepreneurial development. Understood in extremis, the entrepreneurial initiative consists in taking ownership of a part of the natural world and, in collaboration with other agents willing to impose their will on it, in recreating that part of nature, without taking into account its natural integrity, in order to obtain maximum economic benefits. Once the profit logic becomes the sole motivation for action on nature, environmental conservation will be subordinated to the financial success of the enterprise. The negative consequences of changes in the environment can be transformed into new sources of profit by attracting investments in sustainability, as a premise for future profits [32]. Such pecuniary-utilitarian concepts have been, are and will be attractive. The terrestrial world has already been shared. The vast field beyond the Karman line, the Moon, the planets, the asteroid belt, that astronomically close proximity that is technologically accessible, but also countless other worlds, which we cannot yet touch, remain the object of the desire for possession.

Returning to the distinction of Callicott and Mumford with which we began our discussion, as long as we consider that natural items have no value in themselves, being intended exclusively for human use, we only rely for the protection of their integrity on prudential reasons. The minimization of the destruction of places being explored on other planets is indicated only so that they can be studied in their original state, and the management of orbital pollution is recommended only so that the function of telecommunications or geodetic satellites is not affected, causing damage to their owners. Prospecting or research may indeed affect the integrity of distant landscapes on other planets hostile to human life, but as long as we are aware that we are destroying mere instrumental values, there are not strong enough ethical principles to prevent us from doing so. However, a vision of the extra-terrestrial world as an unlimited thesaurus of resources may relieve current and future investors of any considerations for the protection of these resources in the common interest, regardless of current regulations.

### 4.4. Ethics of the Original Appropriation

A multitude of private entities operate in orbit and their number is constantly growing. We are interested in Locke's argumentation because it can morally sanction the process of exploiting the Earth's orbit, which has become more accessible today than ever as a result of both political will and the sudden rise of space start-ups such as Space X. Non-legal reasons for the fair use of extra-terrestrial resources and the sustainability of the orbit may come from observing the Lockean principle of fair ownership. Ownership of a part of the natural world, which did not previously belong to anyone, is acquired simultaneously with the beginning of its use as a resource [30].

This original act of appropriation has one more condition: any individual can freely acquire a part of a nature still untouched, but he is morally obliged to do so in a fair way

to others. Through hard work, the individual enters into an axiological relationship with that part of the natural world that belongs to him. By turning it into private property, he learns not to waste irrationally those natural goods. The notion of original appropriation describes, from an ethical point of view, how a part of nature can pass from the common property of humanity into private property. As it is declared a global commons, the orbit cannot be held in perpetuity but only used.

Locke's theory is interesting because it refers to the individual rights to those natural goods that we consume according to our needs. We return to a previously noted fact. Simmons also observes that Locke says that individual rights to nature are justified for theological reasons, thus substantiating his liberal ethic of action in the natural environment [33]. The term property indicates the exclusive right over a natural object; however, it does not have a legal meaning, but a moral one. This is because there is a relationship between work and property based on that pre-civilizational state in which people were not yet constrained by external laws. To Locke, the human condition is defined in terms of morality and not in legal terms: it consists in the freedom to dispose of one's own person and property within the limits of what he calls natural law [30].

In Locke's view, all discovered natural entities can become individual property. We resume here the Lockean anthropological account because, as Nash observes, it makes possible the ethical reflection on the nature of property [34]. There is still an important nuance to our debate: even if a certain natural item belongs de facto to the person who made the effort to use it, the natural world, in its entirety, remains the common property of humanity. Someone may have the fruits they pick, but not the tree from which they were picked, the wild boar they hunt, but not the forest in which it was hunted. As Bishop notes, property begins to exist through the effort to make natural items available for individual use. No part of nature can enter into property except by mixing it with human labor [35].

Every virgin nature, which exists as a wilderness, is destined for a multitude of acts of individual appropriation, acts which presuppose the legitimate exercise of fundamental and inalienable human rights: the right to life, liberty and health and, as a natural consequence, the right to a part of the goods of nature. We can then conceive that an appropriation of extra-terrestrial wilderness, through a historical effort similar to countless other colonization actions in the past, would be entirely justified because it would involve adding human value and imprinting rationality to a world in which unlimited abundance of mineral resources awaits the coming of man.

### 4.5. Applying the Lockean Proviso

Before moving on to the application of the Lockean proviso, we consider it useful to mention the concept of enframing. This concept belongs to Heidegger and is mentoned because it captures the anthropological consequences of a historical fact: today's technology gives us the means to instrumentalize the natural world. The notion has a deep ontological meaning: natural items tend to appear to us only as a collection of raw materials, as standing-reserves destined for our use. If nature exists for us as a mere source of energy then, even with the use of sustainable technologies, this kind of radical reductionism is still changing our worldview [36]. With this in mind, it is important to take into consideration that Locke does not give a utilitarian meaning to his notion of original appropriation. It is not a question of simply satisfying immediate needs, but of fulfilling a duty, that of preserving, in an irrational world, the existence of rational beings [35]. Because we are social beings, we can acquire through work exclusive rights only over those parts of nature not yet possessed by someone else. The rest remains common property. Once this condition is met, any new act of appropriation is subject to a series of moral imperatives which, together, constitute the Lockean proviso. No one has the right to own more than they can use or to claim more than they can consume. The individual must be able to transform nature into his private property in such a way as to "remain sufficiently and equally good for others" [30]. If, and only if, all these conditions are met the act of appropriation is

justifiable. Only then ethical principles such as justice, equality, and the safeguarding of the common good will guide human action in the natural world.

This programmatic proviso, designed to morally guide the colonization of America, was intended for emigrants from seventeenth-century Europe. The American continent was, for them, terra incognita. Locke seeks to ethically regulate individual freedom of action in a natural environment untouched by the European man. It should remain relevant in any new similar historical situation both because of the way in which it is formulated and because it raises issues of constant relevance such as intergenerational justice. In this regard, Attfield notes that it could lead to an imperative of conservation of natural resources, according to the principle that present generations must leave behind a world whose capacity for support is not diminished for future generations [9]. Since the orbit is a global commons, any harmful consequences that its exploitation could bring should be known and minimized in the name of safeguarding the future of humanity. That is, in Locke's terms, it should be left just as good for future generations.

Still, it must be said that an in-extremis interpretation of the Lockean proviso leads to the idea that by original appropriation are acquired all the rights necessary for the respective exploitation action to be a financial success, including the right to pollution. In a Keynesian vein, Rothbard even goes so far as to quantify the property rights that an act of ownership confers on us. He uses the term "technological units". Such a unit represents the minimum fraction of nature that must be exploited, despite the environmental risks, in order for the initial investment to be profitable [37]. On the other hand, Liebell argues that Lockean theory can help found a sensitive and effective ecological liberalism because it asserts and limits individual rights by establishing rules of fairness that prohibit the waste and plunder of natural resources [38].

As a speculative way of reasoning relying on analogies, heuristic reasoning allows us to at least approximate, if not rigorously determine, the relevance of particular issues involved in analyzed situations. As Nozick notices, Locke's conditions appear to be interdependent: the economic agent must invest his own work in a part of nature not previously held by anyone, appropriating quantitatively and qualitatively only certain elements so that, on the one hand, he does not harm the needs and interests of others, and on the other hand, does not spoil nature by waste or carelessness [39].

Drawing on the epistemological familiarity of the Lockean proviso, both Rosenberg and Pilchman present scenarios plausible enough to take into consideration. In the near future, private law entities will mine areas of extra-terrestrial planetary surface or intensively exploit various asteroid masses. Because any activity in space, even a robotic one, would involve human labor in conditions of extreme hostility, and the ores concerned would be highly demanded on the market, the risk of a waste of resources so difficult to obtain does not exist. Future acts of appropriation would indeed easily satisfy the first and last condition [10]. Avoiding the harm of the rights of others remains the most difficult goal. Private prospecting and exploitation of extra-terrestrial resources would involve a really prohibitive initial investment, thus becoming the exclusive prerogative of an extremely limited category of companies. As the exploitation rights would belong exclusively to the first entrants, that remaining majority would be put in an absolutely unfair situation. Referring to the asteroid belt mining projects, Pilchman argues that the logic of the original appropriation allows for the formulation of moral objections against the permissibility of a legally unrestricted exploitation of the extra-terrestrial environment [10].

Given the heuristic advantages of the imaginary case method, let us now try to answer this question: referring to the Lockean proviso, under what conditions could we possess an orbital altitude?

The extra-terrestrial space, including the Earth's orbit, is declared a common heritage of all mankind. Therefore, every individual has the right to convert this common property into individual property through work, provided that the part of nature under scrutiny, in our case that level of altitude, is not already someone's property. Given the vastness of orbital space, although at present a relatively large number of private law entities profitably

exploit the orbit, we can consider that this condition is and will remain easy to fulfil in the foreseeable future.

The second condition would be to limit ourselves strictly to those natural resources that we need and not to waste them. Whether it's global connectivity, telecommunications or space tourism, orbiting is tied to a growing market for high-tech services. Due to the extremely high costs involved in this type of economic activity, and the fact that exploiting an orbital altitude involves minor to irrelevant changes in the environment being similar to the transit of a ship through international waters, there is no question of waste. This condition is fulfilled.

The third condition remains the most relevant: in order for the acts of appropriation not to harm the vital interests of other human individuals, we should restrict ourselves so as not to harm them now or in the future. This raises the issue of observing the principle of equal access to resources of the same quality, but especially the issue of sustainability of the orbital environment. That is why, given that there are currently no plans to clean the technological waste out of orbit, adopting the conservation of the orbital environment as an ethical principle could help at least formulate a more environmentally responsible liberalism, as part of a long-term agenda of natural resources exploitation in the vicinity of the Earth.

Having in mind only the environmental risks we impose on future generations for justifiable reasons [40], as far as particular cases are alike, such standards of conduct should remain relevant both from an ethical and a legal point of view.

## 5. Conclusions

The present research aims for a better understanding of ethical issues related to orbital environment sustainability as a problem of intergenerational justice. Moral reasoning already has a quantifiable impact on political decisions, helping to legitimize medium and long-term environmental strategies. The same need for a proactive way of thinking that leads to the anticipatory enforcement of Pigouvian taxes, makes moral intuitions regarding pro-environmental behavior become more relevant. Tackling orbital environmental issues is largely a matter of responsible anticipation, hence the relevance of the imaginary case method.

By applying the Lockean proviso we have come to the conclusion that paying more attention to the ethical aspects of the exploration and exploitation of outer space can make a significant contribution to understanding and enriching legal concepts which are still insufficiently precise and which are currently used in formulating regulations on the exploitation of Earth's orbit and even on the moral basis of human activity in other planetary environments that are being explored.

Because private initiative in outer space remains largely uncontrollable legally, the orbital environment can become a completely unrestricted area of freedom of action. Given that there are currently no plans to clean the technological waste out of orbit, adopting the conservation of the orbital environment as an ethical principle could help to formulate a more environmentally responsible liberalism, as part of a long-term agenda of exploitation in the vicinity of the Earth.

Since the orbit remains a global commons, any detrimental consequences for future generations of its exploitation should be known in order to be minimized. Although Locke elaborated his proviso in a specific historical situation, the colonization of the North American continent in the seventeenth century, the appropriation of new resources is still dictated by the same anthropological imperative: the improvement of the human condition. Thanks to this, the original common property over nature can be legitimately transformed into private property at any time. Therefore, even if ecological reasons for the protection of natural systems are not sufficient to lead to the cessation of any activities of destructive exploitation of the orbit, which can always be exceeded by financial considerations, their engagement in an anticipated ethical debate may, at least, prevent a dangerous form

of what is called laissez-faire capitalism. This proves once again the topicality of the Lockean proviso.

**Funding:** This research received no external funding.

**Institutional Review Board Statement:** Not applicable.

**Informed Consent Statement:** Not applicable.

**Conflicts of Interest:** The author declare no conflict of interest.

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
