# Peer review of "The Lockean Proviso and Orbital Sustainability—An Anthropological View"

_sustainability, doi:10.3390/su14073909_

Round 1

Reviewer 1 Report

The paper is interesting and, at the same time, quite peculiar. 

It is very experimental, both in its 'imaginary' case study and in its format. 

The English language is clear enough, but it is not completely up to academic standards and should be revised and improved with the help of a native-speaker. 

This Reviewer does not know if a paper like this is suitable to a Journal as prestigious as Sustainability (even to a Special Issue, with specific topic, of the Journal), but it surely represents an innovative perspective - it even offers an innovative approach. 

To be considered for publication, nonetheless, the paper needs some work. 

The literature review is 'absent', in the meaning that is is 'scattered' all over the article - the Author should organize it better, possibly in a specific section, after the Introduction, entitled Literature Review, containing all the works cited (and, possibly, more), with some analytical comments and explanations, also for a non-specialized audience. 

The methodology of the paper is understood through the reading of the paper itself, but, also in this case, an appropriate section, possibly entitled Methodology, should be 'there', indicatively after the Literature Review, explaining step-by-step the method applied / implemented by the Author, always with the goal of intrinsic reproducibility. 

All the other contents are relatively good, and they can be considered an ideal Results and Discussion section for the paper itself. 

Both in the Introduction and in the Conclusions, the Author should stress on the relevance of his paper in the related field of studies and should explain a little more comprehensively his research aim (Introduction) and, at the level of summary, how he realized it (Conclusions) - that would make the paper extremely coherent and solid, with the implementation of the other sections suggested above. 

All in all, the paper is interesting and quite unexpected - however, "unexpected", here, means "original" and "innovative". 

Surely it can trigger a debate, some scholars would not agree with the conclusions of the article and with the article in itself, but, exactly because it can ingenerate a possibly fertile discussion, the paper can be worthy of our attention. 

As mentioned, nonetheless, before being considered for publication, some issues, especially at the level of format, have to be solved. 

Thank you very much. 

Reviewer 2 Report

Paper is a very interesting application of philosophy. John Locke's philosophy is interestingly applied to the field of environmental policy. According to the authors, the Earth's orbit is used for commercial purposes. These activities are difficult to control. We certainly appreciate the use of analogy and the involvement of the issue of moral sensitivity. We find the analogy with the river regime interesting. We consider the sustainability of the Earth's orbit to be important from a distant future. We consider the warning that legal agreements on the use of extraterrestrial space are quite outdated and that technology has advanced considerably since then. The protection of the environment in space is, of course, justified, in which it is possible to agree with the authors of the study. Authors do not perceive nature as a source of energy. Perhaps in addition to the philosopher Locke, Heidegger could have served as a source of inspirational ideas with his critique of reducing the whole of society to a source of supplies. The use of Locke in the given context of connection with the anthropocentric view is very well fitted to the application level. I also consider the analysis of Locke's attitudes to be good. It can be seen that the authors studied Locke's texts in detail. We consider the visionary potential of the study to be enormous. What the authors describe may become a reality in the coming decades, perhaps centuries. Challenges and impact Locke's reflections on the natural richness of the cosmos are considered very innovative. The application and analysis of Locke's reservation is well described and analyzed. The authors also work with interpretations and comments concerning Locke's reservation. We consider carefully formulated proposals to be interesting and beneficial.
The conclusions of the study are well formulated. They are consistent with the content of the study.
   The study has a good structure. It is adequately structured. The number of bibliographic references is satisfactory.
   I recommend replacing “Locke’s logic” on page 7 with Locke's argumentation. Logic deals with reasoning and inference.
   I also recommend the authors to mention, at least with some mention, that the philosopher Martin Heidegger also commented on the topic of reducing the totality of energy storage. I think a short paragraph about Heidegger's reflection on this issue would help inform the reader. I consider it highly innovative and high quality.

Round 2

Reviewer 1 Report

The paper has been considerably improved. 

  The Literature Review could be expanded further, but that is not a major concern or requirement. 

  All in all, a good revision. 

  Thank you very much.